# Use of a Living Lab Approach to Implement a Smoke-Free Campus Policy

**DOI:** 10.3390/ijerph20075354

**Published:** 2023-03-31

**Authors:** Martina Mullin, Shane Allwright, David McGrath, Catherine B. Hayes

**Affiliations:** 1College Health, Trinity College Dublin, College Green, D02 PN40 Dublin, Ireland; 2Public Health & Primary Care, Institute of Population Health, Trinity College Dublin, Russell Centre, Tallaght Cross, D24 DH74 Dublin, Ireland

**Keywords:** smoke-free campus policy, tobacco-free campus policy, living lab, action research, policy adherence, policy compliance, university

## Abstract

While universities have increasingly become tobacco-/smoke-free, to our knowledge, no campus has reported 100% policy compliance. Innovative approaches to encourage compliance and ongoing data collection are needed. This paper describes actions undertaken, framed within a Living Lab (LL) approach, to implement smoke-free campus policies in an Irish university. The action research comprised student-collected data on observed smoking on campus to evaluate adherence and compliance, first to a smoke-free zones policy (June 2016–March 2018), and then to a smoke-free campus policy (March 2019–February 2020). From June 2016–February 2020, 2909 smokers were observed. Adherence, defined as the average reduction in number of observed smokers from baseline in May 2016, reduced by 79% from 5.7 to 4.9 . Compliance, defined as the proportion of smokers who complied when reminded of the policy, was 90% (2610/2909). Additional activities included development of a broader health promotion programme; identification of a pattern of ‘social smoking’; and promoting increased awareness of the environmental harms of tobacco. Ongoing policy implementation is essential for smoke-free policies and should include data collection and evaluation. Actions framed within the characteristics of a LL achieved fewer observed smokers. A LL approach is recommended to encourage policy adherence and compliance.

## 1. Introduction

One of the responses to the World Health Organization’s call for a total ban on smoking in public places [1] has been the emergence of tobacco-free policies for university campuses. Policies are defined as smoke-free when smoking is prohibited on campus or tobacco-free when all tobacco products including e-cigarettes are prohibited [2]. In 2017, 16.7% of accredited, degree-granting institutions in the United States (US) had 100% smoke- or tobacco-free protections [2]. The number of tobacco-free campuses has more than tripled between 2011 and 2022 [3]. Prohibition of smoking on campus is important as data have shown that almost half of students start smoking at age 18 years or older, and there is a progression from occasional smoking in undergraduates to daily smoking in post-graduates [4].

While high quality evidence on the effectiveness of university policies that prohibit smoking is limited [5,6,7,8], a 2016 Cochrane review of smoking bans in institutional facilities, which included two universities, found that the bans were associated with a reduction in smoking prevalence (RR = 0.72, 95% CI 0.64–0.80) [7]. Separately, two longitudinal studies [9,10] reported that anti-smoking policies significantly reduced smoking and pro-smoking attitudes over time. They concluded that policies that are more comprehensive and incorporate prevention and cessation programmes produce better results in terms of reducing smoking.

As more campuses become tobacco- and smoke-free, attention is directed towards measuring compliance with these policies. Difficulties in definition and measurement have been identified. One difficulty is the terms ‘adherence’ and ‘compliance’ being used interchangeably in other papers [11,12,13,14,15] to mean the degree to which introducing policies reduces smoking. There is no commonly accepted definition differentiating these terms in relation to smoke-free campuses. Another difficulty is the wide range of measures of adherence/compliance, e.g., direct observation of smoking [11,12,13,14,15,16,17,18], self-reported smoking behaviour [9,10,17], self-reported secondhand smoke exposure [10], intention to smoke [10], number of cigarette butts on campus [13,14,15,16,17,19], rates of sign-up to smoking cessation services [12], and attitudes to tobacco-free policies [9,10,11,20,21]. Counting of cigarette butts has been demonstrated to be a valid measure of compliance by Ickes et al. [16]. They counted observed smokers and cigarette butts using a reliable cigarette butt protocol over a one-year period and found a positive relationship between number of violators observed and number of cigarette butts collected. The current study’s use of the terms adherence and compliance are defined in the Objectives.

The positive effects of tobacco-free policies come despite adherence/compliance remaining a challenge [13,22]. Non or partial compliance with policies has been observed in quantitative [13,14,17,19,21,23,24] and qualitative [11,15] studies. While a staff-led ambassador programme was in place, Ickes et al. [14] counted an average of twopeople smoking per visit at campus smoking blackspots compared with a baseline of five. Harris et al. [18] also counted observed smokers during a week-long intervention to increase compliance. A maximum compliance of 74% from baseline was achieved. Lee et al. [19] counted the number of cigarette butts on 19 community college campuses and found butts on every campus. Clemons et al. [25] identified smoking blackspots and counted butts once a week for 18 weeks. For the two-week period while the policy had been announced but not enacted, student ambassadors and signage were used to remind smokers that the policy change was forthcoming. Once the policy was implemented, people found to be using tobacco products were given a verbal warning and could be forced to attend a counselling session. It was also possible for a formal letter to be entered into the university records of staff or faculty. From baseline to the policy having been announced but not enacted, the average number of butts reduced from 50 to 35, (*p* = 0.02); from announcement period to enactment, the average number of butts was 17 (*p* < 0.001). The authors concluded that smoking decreased when social pressures and punishments were used, but the number of butts observed remained significantly greater than zero.

A number of studies assessed strategies to increase adherence/compliance [14,26,27,28]. Adherence/compliance matters because student smokers on a Canadian campus [11] were initially willing to comply with a policy to restrict smoking but observing others disregard it without consequence altered their attitude and subsequent compliance. Fallin-Bennett et al. [26] analysed semi-structured interviews with 68 key informants representing 16 California universities with strategies to encourage compliance from social approaches to heavily punitive enforcement. They concluded that tobacco control policies should primarily be the responsibility of non-university security channels. A study [14] that piloted training undergraduate ambassadors in a large, tobacco-free campus in the US found that undergraduates felt they “were not taken seriously.” For the full programme, they trained part-time university staff and reported that an ambassador programme was a feasible and potentially effective strategy to increase policy compliance. They highlighted the dearth of evidence on effective programmes to improve compliance. A phenomenological study [27] of 20 student ambassadors and four staff found that ambassadors and violators were positive about the tobacco-free policy in a Montana university, with ambassadors feeling an increase in recognition of their role over time despite mixed feelings about their level of authority. Gatto et al. [28] assessed the effectiveness of a policy with peer enforcement only, one year after its implementation in a university in Florida. They found there was only moderate knowledge of the policy and that the majority of respondents (66.8%) indicated they had not approached violators to inform them of the policy and had no intention of doing so in the future. They concluded that new and innovative evaluation tools are needed so that institution leaders can evaluate the effectiveness of policies.

The experience of a university in Dublin, Ireland, in developing smoke-free campus policies forms the background for this research. In 2013, the University Board granted permission to explore the possibility of becoming a smoke-free campus. A Tobacco Policy Committee which included the authors of this paper was established. Baseline data were collected on smoking prevalence [29] and a year-long consultation with over 10,000 engagements was conducted, consisting of online surveys, town hall meetings, working groups, and an undergraduate Students’ Union vote. The Students’ Union voted not to support the proposed policy [30]. From 2014–2016, there were two unsuccessful attempts to agree to proposals to ban smoking on campus. In 2016, with the support of the undergraduate and postgraduate student unions, the University Board agreed to a policy to pilot three smoke-free zones. The three zones chosen were in areas where students and staff had most often voiced complaints regarding second-hand smoke [31]. It was also agreed to anchor the smoke-free policy within a new, broad health promotion framework, which adopted a whole-university approach and included other health topics such as mental health, healthy eating, and physical activity. The smoke-free zones pilot was implemented from July 2016 to April 2018. Between May 2018 and February 2019, results from the pilot were used to negotiate a smoke-free campus policy. From March 2019 to February 2020, the smoke-free campus was implemented. Throughout this process, ongoing literature reviews on tobacco-free and smoke-free policies were used to support negotiations and to design the policy.

A recent systematic scoping review of 75 studies on tobacco- and smoke-free university campuses in 23 countries concluded that while there was a mature body of literature describing the development of smoke- and tobacco-free policies, future studies should quantify the impact of the bans and focus on process factors that might moderate that impact [5]. The Tobacco Policy Committee decided to conduct action research in order to capture both process factors and policy impact.

As part of our action research, Living Labs (LL) were identified in 2018 as a possible innovative evaluation tool that could enhance adherence and/or compliance and support smoke-free ambassadors. The European Network of Living Labs define LLs as user-centred open innovation ecosystems based on a systematic user co-creation approach, integrating research and innovation processes in real-life communities and settings [32]. Hossain et al. [33] define them as a physical or virtual space in which to solve societal challenges, by bringing together various stakeholders for collaboration and collective ideation. Policy makers use LLs to design, explore, experience, and refine new policies and regulations in real-life settings [34]. Importantly, LLs are facilitated rather than managed, because they do not assume any authority over the individual participants. The Tobacco Policy Committee considered this a potentially useful aspect of the LL approach for implementation of policies by universities that are unable to obtain political support for formal enforcement [26].

Hossain et al. [33] identified eight characteristics of LLs:

Real-life environments in which to experiment;Stakeholders who collaborate;Activities that are facilitated rather than managed;Business models and networks that explore feasibility;Methods, tools and approaches that are relevant to measuring human behaviours;Challenges related to the type of LL;Outcomes, both tangible and intangible;Sustainability of a project’s responsibility to the community in which it operates.

From 2018 onwards, the Committee framed the work within these eight LL characteristics.

### 1.1. Study Aims & Objectives

The overall aim of this study was to evaluate the implementation of smoke-free policies at a university campus using participatory action research and to assess the usefulness of a Living Lab (LL) approach in achieving this.

### 1.2. Objectives

To assess adherence to a smoke-free zones policy and a subsequent smoke-free campus policyby comparing the average numbers of observed smokers at baseline (May 2016) to those observed between July 2016 and February 2020. Adherence was defined as the average reduction in the number of observed smokers from baseline.To assess compliance by analysis of the responses from smokers to smoke-free ambassadors’ requests to comply. Compliance was defined as the proportion of smokers who complied when reminded of the policy.To assess the usefulness of a LL approach by comparing the actions undertaken by the university in proposing and implementing smoke-free policies with Hossein et al.’s [33] eight LL characteristics.

## 2. Materials and Methods

### 2.1. Study Design

A dynamic participatory action research design was used for the study [35].

### 2.2. Setting

The university’s island campus in the centre of Dublin city was the real-life environment for this initiative. Students, staff, and visitors to the university participated in this initiative. Three study sites were selected (Figure 1 and Box 1). They were visible, social areas of the campus that were highlighted during a university-wide consultation [30] as being hotspots where smokers gathered and where exposure to second-hand smoke was a problem.

Box 1Description of study sites.
**Study Site**

**Brief Description**

*Central Social Area*
Area with outdoor seating beside building containing café, library, lecture theatres, offices; green space on which people gather during dry weather; a tourist attraction with approximately 1 million visitors per annum; area in front of post-graduate reading room with portico and steps.
*Nursery*
Area around childcare facility, health centre, and student residences.
*Sport*
Areas outside Sports Centre, faculty offices, lecture theatres, offices, and student residences. 

### 2.3. Study Population

The study population comprised students, staff, and visitors to the campus. In 2021, there were 3491 staff and 18,871 students of whom 1594 were part-time [36]. In 2018, over 1 million people visited a tourist attraction at one of the three study sites (Central Social Area) [37].

### 2.4. Intervention Description

This was a complex, composite intervention which included two Action phases and a Negotiation phase.

Action phase I: From July 2016 to April 2018, a smoke-free zones policy (three smoke-free zones) was implemented on a pilot basis and the following support activities were undertaken:Communications: the policy was communicated to the university via a smoke-free website, email, signs and posters on campus, advertising of new smoke-free ambassador roles, ongoing social media, and on-campus information screen campaigns.Ambassador programme: 13 postgraduate student ambassadors were recruited to facilitate direct observation of smokers.Events: launch events promoted the policy; novelty events raised awareness, e.g., cigarette-shaped pinata events, sports club healthy library events, a healthy messages competition, a “count how many butts are in the jar to win a prize” competition, workshops on why the university was smoke-free, smoking voting bins which invited smokers to dispose of their cigarette butts in column A or B with each butt being counted as a vote for a novelty category [38].Support to quit: stop smoking courses twice a year.Open approach: students and staff were invited to contribute to the initiative via teaching and communications.Monitoring: the Tobacco Policy Committee met twice per year to monitor adherence and compliance and to recommend actions to support the policy.

Negotiation phase: After completion of the pilot, from May 2018–February 2019, the Tobacco Policy Committee negotiated the introduction of a smoke-free campus.

Action phase II: Between March 2019 and February 2020, a smoke-free campus was implemented, i.e., the smoke-free policy was extended to the whole campus with three small exceptions: (i) near a banqueting hall; (ii) near a 24 h library, and (iii) beside the campus bar. Actions described above for the smoke-free zones pilot were resumed.

### 2.5. Data Collection and Process

Data on adherence and compliance were collected between May 2016 and February 2020 (Table 1). Baseline data were collected on one day. Data were collected for two years of smoke-free zones. During the 10-month period of negotiation to become a smoke-free campus, data were not collected, and the policy was not enforced. Data collection for the smoke-free campus commenced in March 2019 and ceased at the end of February 2020 as the university closed on 12th March 2020 due to COVID-19 restrictions.

Ambassadors spent seven minutes per study site counting smokers, respectfully reminding them of the policy and asking them to stop smoking or move to a designated smoking shelter/area. Checks were conducted on weekdays during mornings (10 a.m.–12 p.m.), at lunchtimes (12–2 p.m.), and in afternoons/evenings (2–7 p.m.). Ambassadors varied the times at which checks were done within those timeframes to reduce the ability of smokers to predict when they would be present. Data items comprised date, time, ambassador name, number of observed smokers, zone name, weather conditions, and how smokers responded. The initiative coordinator and smoke-free ambassadors met weekly to review smoking numbers and to develop tobacco and broader health promotion messages.

### 2.6. Outcome Measures

#### Adherence and Compliance

Direct observation of smokers was the method used to measure implementation of the smoke-free policy, with adherence and compliance differentiated. Adherence is the number of smokers who adhered to the policy. As this cannot be measured, for the purposes of this study, adherence was measured indirectly by comparing numbers of observed smokers at different time periods after policy implementation to the numbers observed at baseline. To assess adherence, the average number of people smoking per check was calculated by dividing the total number of observed smokers by the number of checks of campus on any given day.

Compliance was the proportion of smokers who responded positively when asked to comply with the policy. Smokers were categorised as “Complied”, “Did not comply”, or “Did not approach.” When a smoker stopped smoking by leaving the smoke-free area or extinguishing their cigarette before being asked, they were classified as “Complied.” When no smokers were observed, the category “No one smoking” was recorded.

### 2.7. Analysis

#### 2.7.1. Adherence and Compliance

Data were analysed in MS Excel. Adherence analysis included baseline, total and monthly data on number of checks, number of observed smokers, average observed smokers per check, and total and percentage reduction in smoking from baseline. The relationship between number of checks and average number of smokers per check was assessed using Spearman’ rank order correlation for non-normal distributions. Percentages within each compliance category were compared for the different time periods.

#### 2.7.2. Comparison with LL Approach

Actions undertaken by the university in proposing and implementing smoke-free policies and the results of those actions were subjectively compared to Hossein et al.’s [33] description of LL characteristics. Based on that comparison, a subjective assessment of the usefulness of each LL characteristic as a framework for understanding the actions/outcomes was determined.

## 3. Results

A total of 2909 smokers were observed, and 79% (adherence (Table 2, Figure 2) was achieved; on average, 4.5 fewer smokers were observed compared with the baseline number of 5.7 in May 2016. Numbers of observed smokers by study site are presented in Table 3. A compliance of 90% (Table 4) was achieved; 2610 out of 2909 smokers approached put out their cigarettes or moved away when reminded of the policy. Compliance and adherence reduced somewhat after the introduction of a smoke-free campus in March 2019 following ten months without a policy implementation strategy in place but reverted within nine months to previously observed high levels. Actions undertaken were compared to LL characteristics and LL was found to be a useful approach for policy implementation (Table 5).

### 3.1. Adherence to Smoke-Free Policies

In year 1 of the smoke-free zones pilot, 939 checks on observed smokers were conducted and in year 2, there were 747 checks (Table 2). In the first year of the smoke-free campus, slightly fewer checks were conducted at the three study sites because the smoke-free ambassadors monitored six data collections points. However, only the three data collection points which corresponded with the study sites used in the smoke-free zones pilot were included in this study. In Year 1 of the smoke-free zones pilot, numbers of observed smokers reduced by 79% from baseline (Table 2); during year 2, a small additional reduction to 82% from baseline was observed. Following the introduction of a smoke-free campus, the average reduction was slightly lower, at 75%.

#### 3.1.1. Observed Smokers May 2016–February 2020

These trends are represented visually in Figure 2. The figure shows that during the pilot (Action Phase I), the average numbers of observed smokers declined immediately, falling to 1.2 in year 1 (Table 2) and 1.0 in year 2, and remained at low levels each month (range 0.7–1.3).

**Table 2 ijerph-20-05354-t002:** Observed smokers at baseline, during two years of a smoke-free zones pilot and during one year of a smoke-free campus policy.

Date: Action	No. Checks	No. Observed Smokers	Average Observed Smokers Per Check	Average Reduction from Baseline in Observed Smokers	Average % Reduction from Baseline in Observed Smokers
May 2016					
Baseline data	9	51	5.7	n/a	n/a
Action Phase I July 2016–April 2017					
Smoke-free zones pilot year 1	939	1132	1.2	4.5	79%
May 2017–April 2018					
Smoke-free zones pilot year 2	747	772	1.0	4.6	82%
Negotiation Phase May 2018–February 2019	No data taken				
Smoke-free zones pilot complete					
Applying for smoke-free campus					
Action Phase II March 2019–February 2020					
Smoke-free campus year 1	721	1005	1.4	4.3	75%
July 2016–February 2020					
Mean for entire follow-up period	802	970	1.2	4.5	79%

More variability was observed during the smoke-free campus initiative (Action Phase I), from March 2019 to February 2020 (Figure 2). In the first smoke-free campus academic semester, (March and April 2019), the average number of observed smokers per check was around 2.0. In the second semester, (October–December 2019), the average number of observed smokers per check dropped to around 1.5. By the third smoke-free campus semester, (January–February 2020), average observed smokers per check had returned to 0.8 and 0.9, levels similar to those observed during the smoke-free zones pilot. Except for the first two months, the average reduction in observed smokers remained at or greater than 75%.

**Figure 2 ijerph-20-05354-f002:**
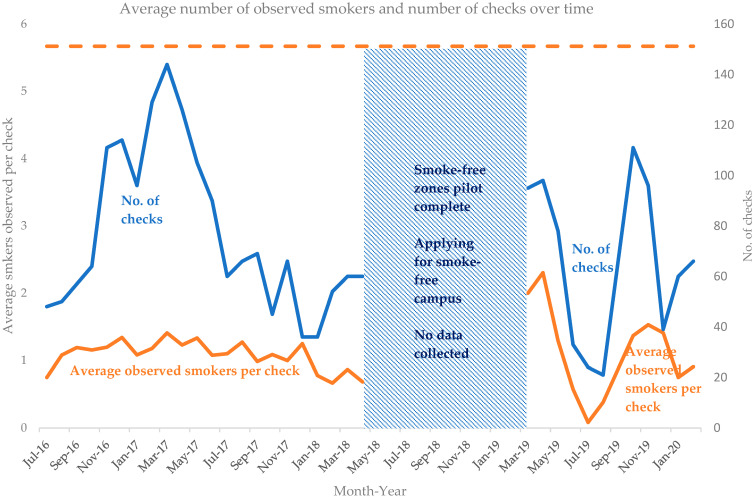
Trends in number of checks per month and in average number of observed smokers per check during the two-year smoke-free zones pilot and one-year smoke-free campus initiative.

During the smoke-free zones pilot, the number of checks varied from 21 to 144, but the average number of observed smokers per check remained at 1 (0.7–1.3), showing a moderate positive correlation (r_s_ (d.f. 20) = 0.58, *p* < 0.001). However, during the smoke-free campus, the average number of smokers per check tended to increase or decrease in parallel with the number of checks (Figure 2), showing a strongly positive correlation (r_s_ (d.f. 9) = 0.92 *p* < 0.001) for this period.

#### 3.1.2. Observed Smokers by Study Site

Throughout the study, much higher numbers of smokers were observed in the Central Social Area site (Table 3). During the pilot, both the Nursery and Sport sites had a more than 90% reduction in average observed smokers relative to baseline. For the duration of the smoke-free campus, however, a slight increase in smoker numbers was observed in the Nursery and Sport sites.

**Table 3 ijerph-20-05354-t003:** Observed smokers at each study site at baseline, during two years of a smoke-free zones policy and during one year of a smoke-free campus policy.

Baseline 13 May 2016	No. Checks	No. Observed Smokers	Average Observed Smokers per Check	Average Reduction from Baseline in Observed Smokers	Average % Reduction from Baseline in Observed Smokers
Central Social Area	3	38	12.7	n/a	n/a
Nursery	3	3	1.0	n/a	n/a
Sport	3	10	3.3	n/a	n/a
Overall	9	51	5.7	n/a	n/a
Action Phase I Smoke-free zones pilot year 1	No. checks	No. observed smokers	Average observed smokers per check	Average reduction from baseline in observed smokers	Average % reduction from baseline in observed smokers
Central Social Area	313	1030	3.3	9.4	74%
Nursery	313	18	0.1	0.9	94%
Sport	313	84	0.3	3.1	92%
Overall July 2016–April 2017	939	1132	1.2	4.5	79%
Smoke-free zones pilot year 2	No. checks	No. observed smokers	Average observed smokers per check	Average reduction from baseline in observed smokers	Average % reduction from baseline in observed smokers
Central Social Area	249	710	2.9	9.8	77%
Nursery	249	23	0.1	0.9	91%
Sport	249	39	0.2	3.2	95%
Overall May 2017–April 2018	747	772	1.0	4.6	82%
Negotiation Phase May 2018–February 2019	No data taken between smoke-free zones				
	pilot and smoke-free campus implementation				
Action Phase II Smoke-free campus year 1	No. checks	No. observed smokers	Average observed smokers per check	Average reduction from baseline in observed smokers	Average % reduction in observed smokers from baseline
Central Social Area	241	823	3.4	9.6	73%
Nursery	240	46	0.2	0.8	81%
Sport	240	136	0.6	2.8	83%
Overall Mar 2019–February 2020	721	1005	1.4	4.3	75%

Footer: Headings and totals presented in shaded areas.

### 3.2. Compliance Amongst Smokers Reminded of the Policy

On 2909 occasions, when smokers were approached and reminded of the smoke-free policy, 90% complied (Table 4). Compliance was highest during the smoke-free zones pilot year 2 at 96% but somewhat lower at 83% post introduction of a smoke-free campus. Compliance with the policy was generally above 90% except in March 2019 (Figure 3), directly after becoming a smoke-free campus, when only 42% (79/190) complied. The following month, (April), compliance rose sharply to 80% (181/226) and by the new academic semester in October 2019, compliance had returned to 95% (144/152), similar to during the smoke-free zones pilots. Figure 3 shows compliance data by month with some of the adherence data from Figure 2.

**Table 4 ijerph-20-05354-t004:** Number and percentage of smokers who complied when reminded of the smoke-free policy by time period.

Date: Action	Complied (n)	Did Not Comply (n)	Did Not Approach (n)	Total (n)	Complied (%)
Action Phase I July 2016–April 2017					
Smoke-free zones pilot year 1	1033	93	6	1132	91%
May 2017–April 2018					
Smoke-free zones pilot year 2	743	29	0	772	96%
Negotiation Phase May 2018–February 2019	No data taken				
Smoke-free zones pilot complete					
Applying for smoke-free campus					
Action Phase II Mar 2019–February 2020					
Smoke-free campus year 1	834	165	6	1005	83%
Overall July 2016–February 2020	2610	287	12	2909	90%

### 3.3. Adherence versus Compliance

Average observed smokers per check (adherence) ranged between 1 and 2 for most of the initiative, and compliance was 84% or higher throughout the four-year period (Figure 3). After the ten months during which no data were collected, with the resumption of data collection for the introduction of the smoke-free campus, lower adherence and lower compliance were observed in March and April 2019. As the smoke-free campus continued, adherence and compliance both increased and by December 2019 had reverted to levels observed during the smoke-free zones pilot.

**Figure 3 ijerph-20-05354-f003:**
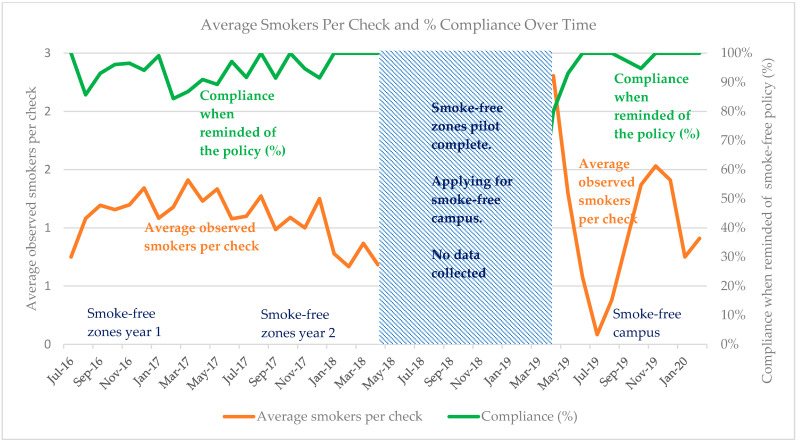
Monthly trends in average observed smokers per check (adherence) and percent compliance when reminded of the policy.

### 3.4. Comparison of Living Lab (LL) Characteristics to Actions Undertaken

It was possible to compare each LL characteristic with actions or outcomes of this smoke-free initiative as the university offered physical, organisational, and human resources in a setting that could usefully be compared to the LL real-life environment and stakeholder characteristics (Table 5). Actions undertaken during the initiative such as requesting rather than enforcing adherence and/or compliance and incorporating activities into ongoing student and staff activities fitted within the following characteristics: activities; business models and networks; and methods, tools, and approaches. As anticipated by the LL approach, challenges were experienced such as non-adherence and new students’ union officers each year who offered different levels of support. Furthermore, as anticipated under the LL approach, this initiative experienced positive outcomes, including fewer observed smokers, development of a broader health promotion initiative, and the identification of a pattern of ‘social smoking’. Finally, sustainability themes that arose by facilitating student actions such as student-led environmental campaigns compared usefully to the LL’s sustainability characteristic.

**Table 5 ijerph-20-05354-t005:** Examples of actions taken by the university relevant to Hossein et al.’s [33] LL characteristics, with an assessment of the usefulness of each LL characteristic to this university.

LL Characteristic Description from Hossein et al. [33]	Examples of Actions Taken by This University, and/or Outcomes of Those Actions Relevant to LL Characteristic	Was the Characteristic of the LL Useful to This University?
1. *Real-life environments* in which to experiment, develop, co-create, validate, and test existing products, services and systems, as well as develop new products and services with stakeholders.	In 2013, before changes were made to the physical space, baseline data were collected on smoking prevalence [29]. About 19% of students smoked; 12% occasionally (non-daily) and 7% daily. In 2014, after a year-long consultation on becoming smoke-free, support for a smoke-free campus was 56%, *n* = 867, amongst undergraduates, 71%, *n* = 199, amongst post-graduates, and 76%, *n* = 427, amongst staff [30]. The Students’ Union did not support becoming smoke-free. Two years of negotiations between the Tobacco Policy Committee and the Students’ Union resulted in a smoke-free zones pilot from July 2016 to May 2018. A smoke-free campus was established in May 2019.	Yes. The university campus offered a real-life environment with physical, organisational, and human resources that could be utilised to develop and test policy innovation.
2. *Stakeholders* who collaborate and may be drawn from business, research and education, public administration, civil society/users.	In 2013, a Tobacco Policy Committee was established with representation from the university Health Service, School of Medicine, Communications, Registrar, College Secretary, Chair of the Group of Unions, Human Resources, Students’ Union, Graduate Students’ Union, School of Dental Science, Safety Office, and Student Ambassadors.	Yes. The policies could not have been implemented without contributions from stakeholders across the University.
3. *Activities* that are facilitated rather than managed because they do not assume any authority over the individual participants, and they are considered an ongoing business activity.	*Communications:* From 2016–2020, all-university emails were sent at the beginning of each academic year to remind of smoke-free policy, advertise stop smoking courses, and recruit ambassadors. In 2016/2019, 23 signs and 14 posters were installed on campus. From 2016–2020, posts were uploaded to campus screens and social media five times per year, e.g., [39,40,41,42,43,44]. The 2019 launch was reported in the media [45,46]. *Smoker interactions:* From 2016–2020, 2909 smokers received face-to-face reminders of the policy from smoke-free ambassadors. Twenty smokers per annum attended stop smoking courses. Smoke-free ambassadors were trained each year from 2016 to 2020 (14 in total). *Events:* 100 people attended a 2018 policy promotion event with a cigarette-shaped piñata [47]. >1000 people entered a 2018 “count how many butts are in the jar to win a prize” competition [48,49]. 415 viewed an online comedy debate about the policy in 2021 [50]. From 2016 to 2019, eight Healthy Library events were hosted by sports clubs encouraging active breaks rather than cigarette breaks-attended by 150 per year.	Yes. Because the smoke-free policy requested rather than required compliance, it was appropriate for a LL. The LL approach facilitated student and staff ideas for activities to support smoke-free policies and the activities could be incorporated into ongoing university activity.
4. *Business models and networks* that explore the feasibility of a business model of complex solutions in real-life contexts; LLs show various types of business models and network structures.	*Business model:* From 2016–2020, the total initiative cost over €150,000 plus in-kind contributions. In 2016, the University Board allocated €10,800 to install signs and smoking shelters on campus [51]. In 2018, a further €26,000 for signs and shelters was allocated on becoming a smoke-free campus [52]. The ambassador programme ran each year costing approx.€4000. A health promotion officer (MM) [53] coordinated the initiative as approximately half of her role. *Networks:* As well as formal networks like the Tobacco Policy Committee, networks were formed to complete specific tasks, to write publications, or to create student projects.	Yes. By staff and students incorporating their smoke-free activities into ongoing work and study, the LL approach offered a feasible means of implementing smoke-free policies and facilitated networks like the Tobacco Policy Committee and student groups completing tasks.
*5. Methods, tools and approaches* that are relevant to measuring human behaviours and interactions and provide an environment of innovation in which to engage all relevant stakeholders in different phases to co-create value.	*Data Collection:* From 2016–2020, on average 79% adherence was achieved and 90% complied when reminded of the policy. *Publications:* In 2017 and 2021, respectively, data on baseline prevalence of smoking [29] and social smoking amongst the student population [4] were published. *Embedding in the Curriculum:* Projects completed by students as part of their coursework included: “No if or butts campaign” in 2019 [54] aimed at reducing cigarette butt waste (business students); attitudes to social smoking [4] (medical students); from 2018–2020, social media and social marketing campaigns on smoking (medical and social marketing students) [44]; guest lectures on delivering the policy were delivered to over 100 students per year (psychology students).	Yes. The collection of quantitative and qualitative data using smoke-free ambassadors was relevant to measuring smoking behaviour. Furthermore, asking staff and students to incorporate the smoke-free policy into their teaching, research, work, or study engaged relevant stakeholders in different phases to co-create value.
6. *Challenges* related to the type of LL and the context in which it operates include temporality, governance and the sustainability and scalability of innovation activities.	Non-adherence to the policy was consistent throughout the initiative. While 79% (see Table 2) adhered compared with baseline, 21% did not. Temporality was a challenge. The election of new students’ union officers each year brought different levels of support. The different competencies and interests of stakeholders were challenging for governance. Scaling the initiative was also an issue. It was not always possible to take on all new ideas brought to the initiative by students and staff.	Yes. The LL approach’s anticipation of challenges is a useful means of setting expectations.
7. *Outcomes*, both tangible and intangible. Tangible outcomes include designs, products, prototypes, solutions and systems, whereas intangible outcomes include concepts, ideas, intellectual property rights, knowledge, and services.	Tangible outcomes included the development of the LL approach described in this paper and, in particular, the development of a broader health promotion initiative in the university [55] with >100 partners working in groups on nine health topics: tobacco, sexual and reproductive health, mental health, healthy eating, alcohol and drugs, physical activity, workplace well-being, breastfeeding, and smarter travel [56]. Each working group uses the LL approach [57]. An intangible outcome was the discovery of high levels of social smoking in the university [4]. Another intangible outcome was the beginning of a better understanding of how students view smoking, as evidenced in the student ‘No Ifs or Butts’ campaign [54] which suggested that students may be more open to environmental anti-smoking messages than health ones.	Yes. This initiative achieved both tangible and intangible outcomes, as expected in a LL.
8. *Sustainability* refers to a project’s responsibility to the community among which it operates. Sustainable innovation and living labs are closely related to each other.	*Sustainability:* The ‘No ifs or butts’ campaign [54] focused on messaging related to the environmental harm of cigarette production and waste as did a campaign by medical students that asked if the environment is a “victim” of tobacco [58]. Cigarette butt waste was also a focus of the smoking voting bins which allowed smokers to vote on novelty categories by disposing of their cigarette butt in column A or B [38], ongoing social media and on-campus campaigns, e.g., [43,59,60] and in an on-campus #ButtVase [48,49] competition.	Yes. Sustainability arose as a theme by facilitating student actions in the LL.

## 4. Discussion

This study sought to determine adherence and compliance with a smoke-free policy in a university and to subjectively assess the usefulness of a LL approach in doing so. Reductions in observed smoking between 75–80% were recorded from May 2016 to February 2020. These data are consistent with international data on policies that restrict smoking [9,10,20]. Nine out of ten smokers complied with the policy when reminded. Both adherence and compliance fell substantially following a period of absence of policy implementation, highlighting the importance of ongoing enforcement to ensure sustainability, as recommended in other universities [6,7,21,23]. Actions or outcomes of this initiative were usefully compared with LL characteristics and LL offers a promising approach to responding to calls for innovative strategies in implementing smoke-free campuses [28]. This initiative achieved strong outcomes: fewer observed smokers; development of a broader health promotion programme; identification of a pattern of ‘social smoking’; and the resonance with students of the environmental harms of tobacco.

The adherence and compliance achieved are encouraging. Amongst young people, the most frequently cited reason for starting to smoke is the influence of friends [61]. As experienced elsewhere [19], the smoke-free policies provided a healthier and cleaner environment for smokers and non-smokers alike and reduced the visibility of smoking on campus.

As in other studies [13,14,19,21,23,24], non-adherence was ongoing over the four years of data collection, particularly in the Central Social Area where average non-adherence was 25%. Brown et al.’s longitudinal study [62] of indoor smoking policy effects on smoking norms concluded that although the policies were implemented amid controversy, support for them increased substantially as effectiveness was demonstrated. Braverman et al. [63] found that university smoke-free policies become more acceptable over time. Our data showed that although there are those who continue to smoke, the majority adhere to the policies over time.

Other universities [6,7,21,23] recommend smoke-free policy implementation strategies and being able to compare patterns of observed smokers highlights the importance of data collection as part of any strategy. Furthermore, the stronger correlation between the average number of smokers per check and the number of checks during the smoke-free campus suggests that more frequent checks may be required when adherence and compliance are reduced as they were in March and April 2019.

However, it is not clear how many checks are optimal to encourage adherence and compliance. Nor is the optimal check duration clear. Ambassador programmes in other universities incorporated checks and durations that varied in length from five minutes to over 50 [14], and from checks one day per week of an unspecified duration [25] to checks three days per week for one hour per day [27]. In this initiative, at least 12 checks per week were completed during term and each check was seven minutes long because that is what resources permitted. Although the Tobacco Policy Committee regularly debated whether smoke-free ambassadors should always spend seven minutes at each survey site or if they should spend more time at sites with lower adherence (i.e., in the Central Social Area) and less in the areas with higher adherence (i.e., Nursery and Sport), the duration of checks was maintained at seven minutes per check, with the number of checks being increased when resources permitted. While maintaining consistent check duration facilitated the delivery of data as presented in this paper, research is needed to assess optimal check duration.

Smoke-free ambassadors did not ask if people had been reminded of the policy previously. An Australian study that asked smokers why they did not adhere to their university’s smoke-free policy [15] identified five themes: defiance against the policy’s perceived threat to self-governance; inconvenience to travel off campus to smoke; smoking as a physiological necessity; unintentional noncompliance through unawareness or confusion of policy boundaries; and ease of avoidance of detection or of exposing others to cigarette smoke. A similar study may be warranted in this university. It may also be useful to draw on literature on adherence/compliance with mask-wearing mandates during COVID-19 to assess other factors that might encourage adherence/compliance. For example, a 2021 US study [64] examined a sample of 233 U.S. residents to determine the role of psychological traits in social distancing compliance and mask-wearing and found that people with more liberal political ideology, who were more risk aware, had higher self-control, and a higher need for cognition, practiced more social distancing and mask-wearing. These behavioral traits are also likely to apply to compliance with tobacco free policies and need futher research. 

Based on the comparison of actions taken by this university, and/or outcomes of those actions, with Hossein et al.’s [33] eight LL characteristics, we conclude that LL is a useful approach to achieving smoke-free policy adherence and/or compliance and offers a response to Gatto et al.’s [28] call for innovative strategies. No papers were found referring to other university campuses as LLs for smoke-free policies. A 2020 systematic review of LL literature in the social sciences that aimed to review applications of LL approaches, identified guidelines for methodological robustness [65]. Although these guidelines are designed for public administration, most of the recommended actions have been undertaken in this initiative, e.g., data collection; using mixed methods to collect information; engaging different researchers in the collection of data; and explicating to what extent a study adheres to various elements of the LL definition. An example of an action not undertaken in this initiative recommended by Dekker et al. [65] is applying process tracing to analyse the effects of interventions, for example, by assessing the effect of communications campaigns on adherence and compliance data. This will be considered for the future. Dekker et al. [65] also note that LLs have developed as a distinctive research and design methodology for co-creating innovation in a real-life context placing emphasis on iterative ways of learning by doing. Their summary reflects the experience of this university. Although not introduced at the outset, using the eight LL characteristics provided a framework in which to develop smoke-free policy implementation. That framework gave a legitimacy to our work which, in turn, gave confidence to use a similar approach to co-develop with partners a whole-university health promotion initiative [56]. The use of a LL approach to implement smoke-free university policies warrants further research. 

The student ambassador model is one which is transferrable to similar third level settings. There are, however, challenges to be overcome before full scale implementation is achieved. One of the biggest challenges is sustainability, as it is well-acknowledged that all interventions experience a ‘voltage drop’ over time [66]. Ongoing commitment and resources are needed to mitigate this and to place the model on a sustainable footing to ensure its ongoing viability. In addition to continuing the actions outlined in Table 5, additional resources to ensure fidelity to data collection are needed, e.g., independent direct observation of the ambassadors’ work is needed for ongoing effective implementation and scale up. Ireland has a commitment to becoming smoke-free by 2030. Government legislation to compel universities to become smoke-free would be a hugely important measure towards achieving this goal. Actions similar to those outlined in Table 5 (Section 1, Section 2 and Section 3) and data collection as described in Table 5 (Section 5) will be continued. To upscale the initiative in the future, increased funding would facilitate more ambassadors and signage on campus and more promotional events and ongoing communications about the health and environmental benefits of being a smoke-free campus. An exploration of the effectiveness of fines on improving adherence is warranted. For other institutions, similar actions and data collection as described in Table 5 are recommended, non-adherence should be expected and planned for, and funding is required to sustain both activities and data collection to support the policy and ensure viability.

Through senior management support, this initiative has been viable because the university invested more than €150,000 from 2016–2020, plus in-kind contributions by staff and students. Although a national plan for a tobacco-free Ireland [67] is in place, that plan has not, as yet, allocated funding to universities of the magnitude provided by the university. It is hoped that the publication of this paper will encourage the development of a sustainable funding model and align with the national policy goal to undertake targeted approaches for young people [67]. This would also respond to the national Healthy Campus Charter [68] call to “Identify and act on opportunities to integrate health and wellbeing into the teaching and learning, research and services of all departments”.

As this initiative has achieved strong outcomes such as the development of a broader health promotion initiative in the university [56] and identification of a prevalence of 21% occasional social smoking on campus [4], it is likely that such investment would be worthwhile. Furthermore, the open approach of the LL has facilitated an awareness of the resonance with students of messaging around the environmental harms of tobacco (8. Sustainability). As universities increasingly respond to Article 12 of the Paris Agreement [69], calling for parties to cooperate in taking measures to enhance climate change education and public participation, it is encouraging that the open approach of the LL facilitated a student-led response to smoking that aligns with the Paris Agreement’s call.

## 5. Strengths and Limitations

To our knowledge, this is the first study to show some evidence of patterns of adherence to smoke-free university policies over time, to differentiate between adherence and compliance, and to demonstrate that a LL approach was useful for implementing smoke-free policies. Future LL smoke-free research could apply all the recommendations from Dekker et al. [65]. The use of participatory action research was a strength of this paper as it facilitated engagement with policy implementation by students and staff.

The assessment of the usability of the LL approach to assess implementation of a smoke-free campus policy was based on subjective opinions. The short period of baseline data collection is an important limitation, and we acknowledge that a longer period would have increased the robustness of the data. However, the authors have no reason to believe that this data collection period was atypical for the pre-smoke-free campus policy period. Non-collection of data on observed smokers during the period when an application to become a smoke-free campus was being made was a missed opportunity; however, this facilitated the assessment of patterns of adherence and compliance to the campus-wide smoke-free policy following a period without an implementation strategy. A further limitation is that ambassadors were trusted to self-report data, and they may have been biased in their reporting. Direct observation of ambassadors was not feasible within available resources. The importance of accurate data was emphasised during the three-week training period. The data ambassadors collected were reviewed on a weekly basis. Thirteen ambassadors had been trained over four years and reported similar patterns of smoking despite having no sight of the previous year’s data, suggesting that reporting bias was minimal. A final limitation is that smokers may have timed their smoking to avoid ambassadors. This risk was mitigated by ambassadors doing checks at varying times within set time periods.

## 6. Conclusions

The smoke-free policies reduced numbers of observed smokers by 79% and achieved compliance of 90%. Taking baseline data on observed smoking and ongoing monitoring of adherence and compliance is recommended for other universities to show the effectiveness of smoke-free policies. Non-adherence, despite constant communications about the policy, was ongoing and should be expected, measured, and prepared for through a well-defined implementation strategy that incorporates data collection.

The LL approach was found to be useful for ongoing implementation of smoke-free policies and offers a promising method of responding to calls for innovative policy implementation methods in other universities. Future research could apply process tracing to analyse the effects of interventions, e.g., assessing the effect of communications campaigns on adherence and compliance. The LL approach achieved positive outcomes including the development of an impactful health promotion initiative across the university, the identification of occasional social smoking as a pattern of smoking not measured by national smoking statistics, and the identification of the resonance with students of environmental messaging about smoking rather than health messaging. Future research should assess the generalisability of LLs to other universities and identify funding models to achieve viability.

## Figures and Tables

**Figure 1 ijerph-20-05354-f001:**
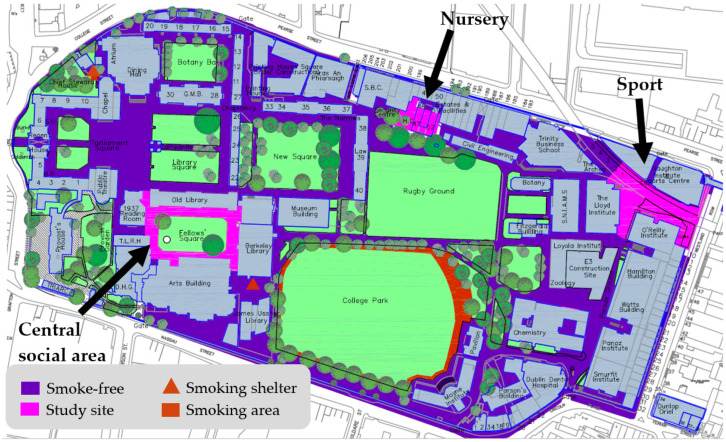
Campus map showing study sites at which baseline and compliance data were collected over a four-year period.

**Table 1 ijerph-20-05354-t001:** Phases of data collection from May 2016–February 2020.

Date	Phase	Description
May 2016	Baseline data	Data collected in three study sites pre-policy initiation.
Action Phase I July 2016–April 2017	Smoke-free zones pilot year 1	Pilot policy initiated and weekly data collection in the three study sites commenced.
May 2017–April 2018	Smoke-free zones pilot year 2	Weekly data collection continued.
Negotiation Phase May 2018–February 2019	Pilot complete. No data collected.	Negotiating for smoke-free campus.
Action Phase II March 2019–February 2020	Smoke-free campus year 1	Smoke-free campus policy initiated, and weekly data collection resumed.

## Data Availability

Anonymised data available from Martina Mullin on reasonable request.

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
