# Peer review of "Use of a Living Lab Approach to Implement a Smoke-Free Campus Policy"

_ijerph, 2023, doi:10.3390/ijerph20075354_

Round 1

Reviewer 1 Report (Previous Reviewer 2)

Dear authors,

Thank you for conducting the study. It is essential to continue developing literature regarding implementation strategies for smoke-free policies in different settings. The article reads well, and the authors have addressed previous comments. 

Author Response

Reviewer 2 Report (New Reviewer)

Comments in relation to the many changes made to the document: I thank the authors for the effort in making the modifications in the different sections of the manuscript, which in my opinion have substantially improved the entire writing of the study carried out. Even so, some aspects of the material and methods section (page 6) are not clear in relation to: 1. Action Phase 2: The authors state: Ambassadors spent seven minutes per study site counting smokers, respectfully reminding them of the policy and asking them to stop smoking or move to a designated smoking shelter/area. Checks were conducted on weekdays at lunchtime (12-2pm), after noons/evening (2-7pm) and mornings (10am-12pm), etc. I have doubts if there is specific information regarding these observation schedules, this could influence the compliance rates. What were the reasons for doing them at the same times always? The community could avoid smoking during these observations and continue smoking at other times without observation. They could provide some explanation and comment it on limitations section more extensively.

Finally, congratulate the researchers for advancing smoke-free spaces at the university where there is a large population of young people!

Author Response

Reviewer 3 Report (New Reviewer)

This is revised verion reviewed by a new reviewer. Glad that annotated version was provided to review. The action research whose results are presented like and quasi-experimental is appropriate as there was not more information that was collected to assess what factors other than the intervention would have likely confounded the changes observed. Perhaps this may even be out of scope for this type of research. limitations are acknowledged the authors, but what biases by the those gathered would have affected the results and how this intervention can be upscaled in similar settings and what challenges would have be overcome should also be acknowledged in discussion. Also a note on sustainability of effects of intervention be discussed, as the effects after implementation are likely to wane over time.

Author Response

Reviewer 4 Report (New Reviewer)

Dear authors,

Thank you for this study describing a LL approach for a smoke-free campus with encouraging results in terms of impact on the smokers by reducing their number and making students aware of the environmental harm and risks on health (even if this latter was less impactful) due to smoking.

I ask for some clarifications throughout the text:

Lines 350-351: then explain what is the difference between adherence and compliance.

Lines 355-56: could you explain why the number of smokers relative to cigarette butts is a valid measure of compliance? How do you measure the number of smokers counting the cigarette butts?

Line 376-377: if there is difference between the two terms, why do you report them in this manner Adherence/compliance that seems they are really interchangeable.

Line 942: adherence and compliance or adherence/compliance? Not clear if they have a different meaning; you studied the two concepts separately or they are interchangeable?

Line 1079: what are the three small exceptions?

Line 1265:  I suppose that mornings (from 10 am to 12 pm) should be put for first and then follow the chronological order.

Line 1271: here you define what is adherence and compliance but you mention them much earlier in the text without defining them.

Line 1332: could please explain better these values between brackets (4.5/5.7)

Figure 2: What is the character after the first numbers of the Month year?

I general I suggest to write a few lines about the sustainability of this LL approach in terms of:

Which activities/checks are to be periodically realized to ensure the smoke free policy campus on the long term?

How to continue the activities/checks to ensure the adherence to a smoke-free policy campus on the long-term?

How to maintain stable the smoking reduction?

Which are the resources in terms of personnel and funding to continue these activities?

What do you expect should be done in the universities about tobacco control?

Author Response

This manuscript is a resubmission of an earlier submission. The following is a list of the peer review reports and author responses from that submission.

Round 1

Reviewer 1 Report

The paper adresses a relevant topic for university (and/or school) campuses. The research conducted is sound and written in a very clear manner. Nevertheless, the conclusion could be more detailed to give due consideration to the research carried out. Furthermore, it would be interesting to compare the smoke policies to the mask policies implemented during the pandemic in follow-up research.

Reviewer 2 Report

Please revise the English language of the manuscript - there are grammar mistakes that need to be addressed. 

Overall, the manuscript is poorly presented. The introduction is extensive and there is a lack of transitions throughout the section. Also, It presents inconsistencies and lack of clarity in formulating the purpose and aims of the study, methodological issues including statistical analysis, and in the presentation of the results, compromising the conclusion. 

Reviewer 3 Report

Introduction is well referenced but too extensive, would suggest shortening.

Otherwise, ideas are well presented. The authors differentiate between "adherence" and "compliance" and extensively cite literature, which is a strenght overall.  

Reviewer 4 Report

Dear Authors,

a few comments that could improve your article.

1.       In the introduction, I miss the information why this is important. You show very accurately what others have done, but I miss the narrative line. It's very hard to read, the paragraphs seem completely disconnected, and some of the information is even redundant. Work on the logical sequence, the narrative in the introduction.

2.       Throughout the text, single brackets are left in various places (e.g. opening bracket, no closing bracket), missing periods, strange characters. This gives the impression of sloppiness. Please correct.

3.       I have doubts about "baseline". Compared to other data collection periods, it contains critically little data. What if such a high rate is a coincidence, i.e. there were just 9 times more people smoking? This is why the following indicators are so large... Please, also add why these and not other places were selected for the purposes of research.

4.       And unfortunately I completely don't understand the approach with LL. If you used LL to design the entire research - then this should be mentioned from the beginning of the article and it should definitely look different. In this case, I feel like at the end you want to check "oh, maybe we'll fit into this approach." acc. me definitely not - show what you really meant and that it is relevant to the article. Maybe this part should not be here at all, but should be a separate, completely different article?
